# R-MAE: Regions Meet Masked Autoencoders

Duy-Kien Nguyen[1,2][*]   Vaibhav Aggarwal[1]   Yanghao Li[1]
Martin R. Oswald[2]   Alexander Kirillov[1]   Cees G. M. Snoek[2]   Xinlei Chen[1]

[1]FAIR, Meta AI     [2]University of Amsterdam

## Abstract

In this work, we explore *regions* as a potential visual analogue of words for self-supervised image representation learning. Inspired by Masked Autoencoding (MAE), a generative pre-training baseline, we propose masked region autoencoding to learn from groups of pixels or regions. Specifically, we design an architecture which efficiently addresses the one-to-many mapping between images and regions, while being highly effective especially with high-quality regions. When integrated with MAE, our approach (R-MAE) demonstrates consistent improvements across various pre-training datasets and downstream detection and segmentation benchmarks, with negligible computational overheads. Beyond the quantitative evaluation, our analysis indicates the models pre-trained with masked region autoencoding unlock the potential for interactive segmentation.[1].

## 1  Introduction

There has been a significant progress of self-supervised pre-training in Natural Language Processing (NLP) over a short period of time, showing the potential of pre-trained language models from huge amounts of data. This progress has been mainly brought about two lines of research, the autogressive language model in GPT (Radford et al., 2018; 2019) and the masked language model in BERT (Devlin et al., 2019). While being different in the design of pre-text task, both approaches learn to predict missing words given the available content. Such reconstructive pre-training enables language models to capture complex and long-range context in documents, resulting in a general learner for various NLP downstream tasks (Brown et al., 2020).

Inspired by the reconstructive design of masked language modeling in NLP, recent self-supervised learning approaches in computer vision also propose to directly predict masked patches from visible image content (Peng et al., 2022; He et al., 2022; Xie et al., 2022). Indeed, the idea of masked autoencoding in vision proves its effectiveness in learning visual representations, reaching state-of-the-art performance in image recognition (He et al., 2022; Woo et al., 2023). Among these methods, Masked Autoencoding (MAE) (He et al., 2022) that employs an asymmetric design and a high masking ratio proves to be a simple and powerful vision learner. Notably, Li et al. (2021) show that unlike supervised or contrastive learning, MAE improves the upper bound of object detection and segmentation compared to a long and optimal recipe of training from scratch.

However, for visual understanding, MAE has not yet reached the same performance level as language models. Despite the benefit of learning from unlabeled data, MAE still lags behind in its scalability (Zhai et al., 2022; Dehghani et al., 2023) and other emergent properties (*e.g.*, one that explicitly capture human-relatable segments (Caron et al., 2021)). This may come from the fact that the raw pixel values are *continuous* signals of the visual world, whereas words are *discrete* human creations. Motivated by this, we examine the concept of '*region*' (Girshick et al., 2014) as a potential visual analogue of words for pre-training, as regions offer similarly discrete information about which group of pixels belong together. By learning from regions in the image, the model can hopefully be less biased towards raw pixels and focus more on the *grouping* of pixels that encode parts, objects, and scenes. Thus it can further advance the performance on tasks like object detection and segmentation.

Specifically, we propose 'masked Region Autoencoding' (RAE), as a reconstructive pre-text task to learn from regions. In RAE, each region is represented as a binary region 'map', with each value

---

[*]Work done during an internship at FAIR.
[1]The code is provided at https://github.com/facebookresearch/r-mae

indicating whether a pixel belongs to the current region or not. We can then follow a similar procedure in MAE to learn a *region-aware* representation by predicting masked portions of the input regions.

However, unlike MAE that reconstructs a single input image in its decoder, learning from regions requires our pipeline to efficiently deal with *one-to-many* mappings. This is because a pixel in the image can belong to an unknown number of regions. In addition, different from color channels in pixels that appear in a pre-defined order (*e.g.*, RGB), the reconstruction of multiple regions needs to maintain *permutation equivariance* – a swap of two regions in the input should automatically lead to a swap in the output. To address these challenges, we explore several architecture variants for RAE and converge to a 'length' variant that compresses each spatial region to a single *query* vector. We show our final design is both efficient and effective.

RAE is fully compatible with MAE. When integrated, we name our approach R-MAE, short for Region-aware Masked Autoencoding. Since we use regions which are fully computable from mere images, R-MAE enjoys the same range of applicability as MAE. Empirically, we find R-MAE can generate useful representations for dense vision tasks such as object detection and segmentation, which we thoroughly study with our experiments. Specifically, we highlight:

- RAE *alone* reaches strong performance, especially when fed with high-quality, off-the-shelf regions (Kirillov et al., 2023) – *better* than MAE;

- Even with regions from a simple clustering algorithm (Felzenszwalb & Huttenlocher, 2004), R-MAE offers consistent improvements over MAE on multiple settings, and reaches state-of-the-art performance without compromising pre-training efficiency;

- Qualitative visualizations show our pre-training is indeed more region-aware, or *instance-aware* compared to others;

- As a final demonstration, pre-trained R-MAE models can be potentially used as a promptable, 'interactive segmenter' beyond representation learning.

## 2  RELATED WORK

We first review two *intrinsic properties* of regions, which have driven their popularity:

**Local.** In machine learning algorithms images are typically treated as holistic entities (Krizhevsky et al., 2017; Chen et al., 2020b), but real-world photos have rich spatial structures and local contents can vary across the same scene (Asano et al., 2019). This became a strong motivation for the well-known R-CNN series (Girshick et al., 2014; Girshick, 2015; Ren et al., 2015; He et al., 2017), especially with Region-of-Interest (RoI) operations on local feature maps (Girshick, 2015). The same holds for contrastive or Siamese learning (Chen et al., 2020b; He et al., 2020; Radford et al., 2021; Grill et al., 2020; Chen & He, 2021; Caron et al., 2021), where 2D signals are generally suppressed into global vectors for inter-image contrast. Realizing its potential downside for localization, many follow-up works (Xie et al., 2021c; Pinheiro et al., 2020; Roh et al., 2021; Xiao et al., 2021; Xie et al., 2021a;b; Yang et al., 2021; Gansbeke et al., 2021; Wei et al., 2021; Hénaff et al., 2022) have shifted focus on intra-image contrast, which use features from local geometric entities (*e.g.* points (Wang et al., 2021), regions (Hénaff et al., 2021) or both (Bai et al., 2022)). Meanwhile, reconstructive methods (He et al., 2022; Bao et al., 2022; Wei et al., 2022; Chen et al., 2022b) as denoising autoencoders (Vincent et al., 2008) preserve the 2D structure. It is therefore unclear how regions can further help in this regard.

**Object-centric.** Reconstructive learning is the dominating paradigm in pre-training natural language representations (Devlin et al., 2019; Brown et al., 2020), and while steady progress is made (Chen et al., 2020a; He et al., 2022), computer vision models are still lagging behind. One crucial difference between the two fields is that language consists of semantically meaningful discrete words, while images are raw continuous signals recorded in pixels. Meanwhile, in vision, objects can serve as a natural counterpart to words – they are constantly referred and manipulated as we interact with the visual world (Koffka, 2013; Zhang & Maire, 2020), and they can often be captured, albeit not perfectly, by regions (Uijlings et al., 2013; Arbeláez et al., 2014). By enhancing MAE's region awareness, we hope to uncover novel ways to bridge the gap between vision and language.

Next we discuss how regions are *generated* and *utilized*:

**Source of regions.** Regions can come from various sources (*e.g.* human annotations (Lin et al., 2014), spatial heuristics (Hénaff et al., 2021), clustering/segmentation (Felzenszwalb & Huttenlocher, 2004; Achanta et al., 2010), object proposals (Uijlings et al., 2013; Arbeláez et al., 2014), or motion segmentation (Pathak et al., 2017)). Most recently, the Segment Anything Model (SAM) proposed by Kirillov et al. (2023) stands out as a universal model for generating region proposals. As an initial exploration, our study mainly focuses on pre-computed, clustering-based regions (Felzenszwalb & Huttenlocher, 2004), but we also verify the effectiveness of R-MAE using regions generated from SAM. Moreover, regions can be jointly discovered (Hénaff et al., 2022) or updated (Bai et al., 2022) with representation learning, which is left for future work.

**Use of regions.** There are at least three other ways to leverage regions in MAE. One is to bias the random masking strategy (Li et al., 2022a), which is less general and can be sensitive to region qualities (Li et al., 2022a). Second is to revisit the RoI operation (Ren et al., 2015) and contrastive learning, which is costly with Siamese encoders (He et al., 2020; Chen & He, 2021), and has been extensively studied (Hénaff et al., 2021; Xiao et al., 2021; Xie et al., 2021b; Wei et al., 2021) even with MAE (Zhou et al., 2022). Third is to view regions as an extra *modality*, and treat the task as a multi-modal learning one (*e.g.* with text (Geng et al., 2022; Singh et al., 2022) or a depth map (Bachmann et al., 2022)). This is closest to our work, yet the lightweight design of R-MAE makes it especially well-suited to learn representations using regions.

## 3 APPROACH

**Background on Masked Autoencoding.** Since Masked Autoencoding (MAE) (He et al., 2022) is the foundation and baseline of our approach, we first summarize it as background knowledge. As the name suggests, MAE uniformly masks out a portion of an image and learns to reconstruct by directly predicting raw pixel values. To provide a meaningful and challenging task for images, a high mask ratio $\beta_I$ (*e.g.* 75%) is used by default. The reconstruction is compared against the ground-truth with a simple $\ell_2$ loss.

As an autoencoder (Vincent et al., 2008), MAE instantiates its encoder and decoder with vision transformers (ViTs) (Dosovitskiy et al., 2020). ViTs directly 'tokenize' images as sequences of patches, which paves the way for MAE's efficient encoder pre-training that *removes* (and not replaces) masked tokens. Given visible tokens from the pixel encoder, the fixed-sized (8-block, 512-dimensional) pixel decoder then reconstruct masked patches via pixel regression. After pre-training, the pixel encoder is transferred as a visual backbone for downstream tasks (Li et al., 2022b).

### 3.1 RAE: MASKED REGION AUTOENCODING

**Region maps.** To perform masked region autoencoding, we first simply follow MAE and prepare them to be 'image-like'. Specifically, each region can be represented by a binary-valued region map similar in size to the image. Each element on the map, with a value of either in 0 or 1, indicates whether the corresponding location belongs to the region or not. Now, given any partially visible region map (mask ratio $\beta_R$), we can ask the model to complete it, the same as MAE does for pixels.

**Architecture.** Similar to MAE, the proposed architecture contains an encoder and decoder for region autoencoding. We follow MAE and simply use ViT blocks (Dosovitskiy et al., 2020) for both. However, just a region encoder-decoder pair is insufficient, as our ultimate goal is to obtain a pre-trained *pixel* encoder. Therefore, we maintain the pixel encoder, and use a *neck* of a single ViT block to match dimensions and (optionally) propagate information before feeding into the region decoder. Such a configuration also makes effective use of the abundant contextual information available in the pixels to pre-train the encoder. See Fig. 1 for an overview.

**One-to-many mapping.** While regions can be considered as an additional *modality* to pixel-based MAE, the problem addressed here presents a distinctive challenge that cannot be fully captured by this view alone. Compared to other modalities (*e.g.* depth or semantic maps (Bachmann et al., 2022)) for which there is a one-to-one correspondence to pixels, the mapping between images and regions is one-to-many: one pixel can belong to an unknown number of regions.

One naïve implementation is to merge the $k$ regions in the *channel* axis. In this way, they can be viewed as a single image, and the computations are shared in the intermediate blocks. But unlike

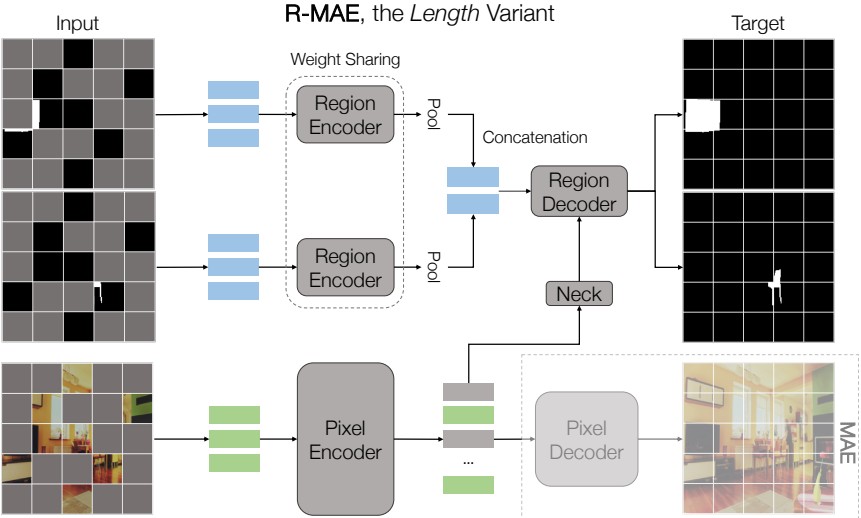

Figure 1: **Region-Aware Masked Autoencoder (R-MAE).** The masked region autoencoding as a standalone task learns to reconstruct multiple region maps in parallel given visible region and image patches. The region encoder generates region embeddings by pooling features from visible region patches. The region decoder then takes region embeddings and decodes them into region maps using image features from the pixel encoder. By treating *regions as queries*, it effectively balances speed and accuracy. The design of our architecture allows its integration with pixel reconstruction in MAE (de-highlighted).

natural images which have fixed channel orders (*e.g.*, RGB), randomly sampled regions can appear in *any* order. It would be ideal if the solution preserves *permutation equivariance*.

Fortunately, this happens to be the very problem encountered in object detection. The mainstream solution, as promoted by R-CNN (Girshick et al., 2014), is to sample and stack regions in the *batch* axis, and process each of them separately. In masked region autoencoding, this means each region map will go through the encoder-decoder in isolation: If there are $b$ images and $k$ regions per image, the network must be applied $b \times k$ times. This is expensive – so how to reduce the cost?

**Regions as queries – the length variant.** Our final idea is inspired by DETR series (Carion et al., 2020; Nguyen et al., 2022), which uses 'object queries' as substrates to decode objects. In a nutshell, each region is first encoded and pooled into a 1D *embedding*; then multiple region embeddings are concatenated along the sequence *length* (Dosovitskiy et al., 2020) axis to form 'region queries'; and finally, these region queries will decode region maps from the output of the pixel encoder (through the neck, see Fig. 1 for details). Since ViT blocks are *set* operations w.r.t. the input (Vaswani et al., 2017), this solution is permutation equivariant by design.

The last decoder block is responsible for expanding region queries *spatially*. Note that because the decoder has two sets of inputs, its blocks follow the three-layer design (Carion et al., 2020), with an extra *cross-attention* layer that uses outputs from the neck to generate keys and values. Different from standard attention layers that

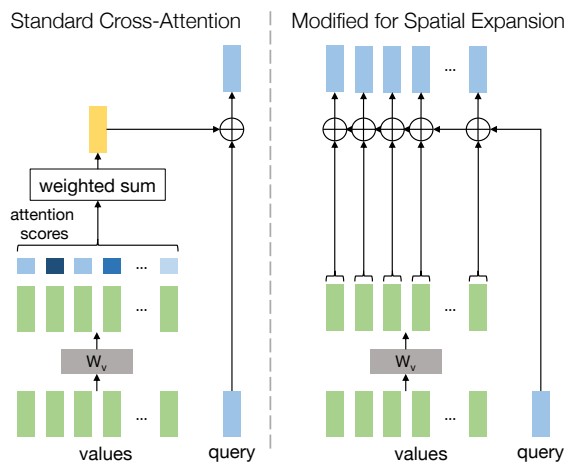

Figure 2: The region query is **spatially expanded** in the *length* variant. We modify the standard cross-attention layer (Carion et al., 2020) (left). Given a region query, it is summed with all value vectors to expand its spatial axes (right). A small MLP head is attached afterwards. This design enables the reconstruction of region maps from the region queries efficiently.

| Query | MoCo v3 | MAE | R-MAE | Query | MoCo v3 | MAE | R-MAE |
|-------|---------|-----|-------|-------|---------|-----|-------|

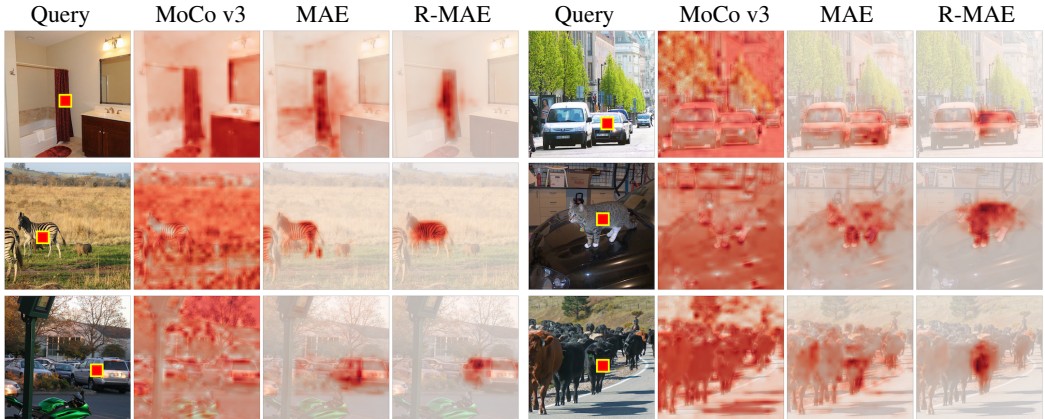

Figure 3: **Attention maps from a Vision Transformer pre-trained with R-MAE**. In each group from left to right we show the original image with the selected query (denoted by red square); three attention maps corresponding to the query generated from i) MoCo v3; ii) MAE; and iii) R-MAE. All methods are pre-trained on COCO `train2017`. In every row from top to bottom, we show 3 types of the query: i) rigid objects, ii) non-rigid objects, iii) multiple objects. Regions with darker red colors in the attention map denote larger attention weights. Compared to the baselines, the attention map from R-MAE is more *instance-aware*.

compute a weighted sum (with keys) over values to produce the output (Fig. 2, left), we expand the query by directly adding it to all the values (Fig. 2, right). A small MLP head is attached afterwards to predict region maps on these spatially expanded features. Since this variant alleviates the linear complexity w.r.t. number of regions $k$, and still maintains the desired property w.r.t. permutation, we choose it as the default for RAE. Since this variant alleviates the linear complexity w.r.t. the number of regions $k$, and still maintains the desired property w.r.t. the permutation, we choose it as the default for masked region autoencoding.

**Loss.** While the $\ell_2$ loss fits real-valued pixel predictions, by default we use the cross-entropy loss for binary-valued regions which is effective for binary classification.

### 3.2 R-MAE: REGIONS MEET MAE

As masked region autoencoding is fully compatible with MAE, they can be trained in conjunction by simply restoring the pixel encoder and applying a joint loss with an equal weight (see Fig. 1). Note that: (i) The pixel branch feeds to the region branch, but *not* vice versa; (ii) The mask is shared between two branches which prevents information leak and creates a more challenging pre-text task. We name this framework R-MAE, short for Region-aware Masked Autoencoding.

Interestingly, Fig. 3 shows that when pre-trained with R-MAE using unsupervised, image-computable region maps (Felzenszwalb & Huttenlocher, 2004), ViT features are shown to be more *instance-aware*. In particular, its attention map focuses more on the objects given the query compared to the reconstructive (MAE (He et al., 2022)) and contrastive (MoCo v3 (Chen et al., 2021)) baselines. The ViT features pre-trained with R-MAE reveal its localization capabilities through the attention map, with strong focus on objects across different locations.

## 4 EXPERIMENTS

### 4.1 EXPERIMENTAL SETUPS

**Source of regions.** By default, we use regions generated from the unsupervised, image-computable Felzenswalb-Huttenlocher (FH) algorithm (Felzenszwalb & Huttenlocher, 2004). It is fast, efficient and covers the whole image that underlies classic object proposal methods (*e.g.* selective search (Uijlings et al., 2013)). The use of FH region maps allows our self-supervised method to inherit the wide applicability on multiple domains. In addition, we also ablate regions from different sources such as, panoptic regions – ground-truth annotations from the COCO dataset (Lin et al., 2014), and regions generated by the SAM model (Kirillov et al., 2023).

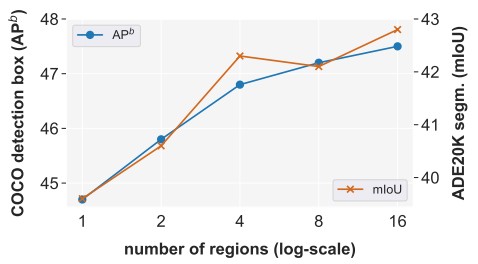

(a) **RAE performance w.r.t. number of regions.**

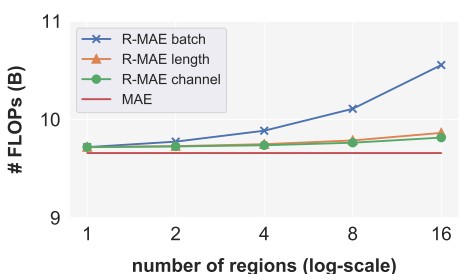

(b) **Complexity of RAE variants in R-MAE.**

| variant | $AP^b$ | $AP^m$ | mIoU |
|---|---|---|---|
| channel | 46.1 | 40.9 | 42.0 |
| batch | **47.2** | **41.8** | **43.3** |
| length | **47.2** | **41.8** | 42.1 |

(c) **RAE variants.**

| source | $AP^b$ | $AP^m$ | mIoU |
|---|---|---|---|
| FH | 47.2 | 41.8 | 42.1 |
| panoptic | 45.9 | 40.7 | 43.4 |
| SAM | **50.6** | **45.1** | **46.8** |
| MAE | 50.1 | 44.6 | 45.9 |

(d) **RAE source of region.**

| strategy | FLOPs | $AP^b$ | $AP^m$ | mIoU |
|---|---|---|---|---|
| MAE | 9.7b | 50.1 | 44.6 | 45.9 |
| RAE $\leftrightarrow$ MAE | 9.8b | 50.3 | 44.8 | 46.6 |
| RAE $\leftarrow$ MAE | 9.8b | **50.6** | **45.0** | **46.8** |
| RAE $\rightarrow$ MAE | 9.8b | 50.3 | 44.7 | 46.2 |

(e) **Integration with MAE in R-MAE.**

Table 1: **Ablation studies** on detection and segmentation. We show: (a) A higher number of regions helps to improve the performance of masked region autoencoding (RAE); (b, c) The *length* variant that treats 'regions as queries' provides the best trade-off between accuracy and complexity among the three variants we have studied; (d) High-quality regions from SAM (Kirillov et al., 2023) contribute to significantly boost the performance of RAE, *better* than MAE; (e) The integration between RAE and MAE in the R-MAE framework, where we find the asymmetric design that only feeds pixels to regions works best. Default settings are shaded in gray .

**Pre-training datasets.** Deviating from prior practices (Bao et al., 2022; He et al., 2022), we develop RAE and R-MAE by pre-training on COCO `train2017` (Lin et al., 2014). This is due to the scene-centric nature of the images in COCO and the presence of ground-truth regions which can serve as useful oracles. Following (Hénaff et al., 2021), FH is run at three scales: $\{500, 1000, 1500\}$, which also set the minimum cluster sizes. Since this dataset (118k images) is significantly smaller than ImageNet (1.4m), we pre-train for 4k epochs instead of 800 (He et al., 2022). For fairness, we also pre-train ViTs with R-MAE on ImageNet (Deng et al., 2009) for 800/1600 epochs. In this case, we extract FH region maps with a single scale of 1000 as in Hénaff et al. (2021).

**Other pre-training details.** Unless otherwise specified, we exactly follow MAE (He et al., 2022) for hyper-parameters. Our base learning rate is set to 1e-4, which offers better stability during training and maintains the baseline performance (see Appendix). The length variant is used. ViT-B (Dosovitskiy et al., 2020) is set as the pixel backbone, and a 1-block, 128-dimensional ViT is used for the neck, the region encoder and the region decoder. A 3-layer MLP acts as the region predictor after the decoder block. $k{=}8$ regions are randomly sampled per image with replacement, and a mask ratio of $\beta_R{=}0.75$. When combined with pixel regression in MAE in R-MAE framework, the pixel branch feeds the region branch, and the random masks are shared.

**Downstream transfers.** The pre-trained vision transformers serve as the backbone for downstream tasks. We simply use the recipe from ViTDet (Li et al., 2022b) for object evaluation on COCO, and report mean Average Precision (AP) for both box detection ($AP^b$) and instance segmentation ($AP^m$). Specifically, the learning rate is linearly warmed up for the first 250 iterations and decayed at 0.9 and 0.95 fractions of the total number of training steps by a factor 10. The input image size is $1024 \times 1024$ with large-scale jitter between a scale range of $[0.1, 2.0]$. We finetune for 100 epochs with batch size of 64. For semantic segmentation, we evaluate on ADE20K and report mean Intersection-over-Union (mIoU) as the main metric following MAE (He et al., 2022) (*e.g.*, run each setting 3 times and take the mean). In sum, all hyper-parameters here are following standard practices for fair comparisons of pre-training settings.

## 4.2 EXPERIMENTAL RESULTS

**Ablation studies.** In Tab. 1, we ablate the most important design choices in RAE and R-MAE: Tab. 1a shows the performance of the RAE *alone* w.r.t. the number of regions. RAE improves when more

regions per image are sampled during pre-training. From Tabs. 1b and 1c, we conclude the *channel* variant is efficient due to the share of computation in the intermediate blocks of the architecture, but lags behind in the performance. This proves that learning *permutation equivariance* of multiple region maps within an image is non-trivial. While the *batch* variant effectively deals with the permutation of regions and demonstrates strong performance, it comes with high computational cost (see Tab. 1b). By treating regions as queries, the *length* variant provides the best trade-off between speed and accuracy, which is important to process multiple regions per image.

In Tab. 1d, we compare the performance of RAE with regions from different sources: FH regions as our default setting, panoptic regions from COCO ground-truth (Lin et al., 2014) and regions generated by SAM (Kirillov et al., 2023). While panoptic regions only improve on semantic segmentation, region maps from SAM contribute to boost the performance of RAE by a *large* margin on all tasks compared to the default FH regions. Surprisingly, RAE alone with SAM regions outperforms MAE (50.6 *vs*. 50.1 for $AP^{box}$ and 46.8 *vs*. 45.9 for mIoU) with less computational requirements (more details in Appendix). This validates that masked region autoencoding is an effective pre-text task especially when fed with high-quality regions.

While SAM regions are superior in accuracy, we still focus on regions generated by FH (Felzenszwalb & Huttenlocher, 2004) – a fast and simple clustering algorithm. Unlike SAM, FH algorithm is fully unsupervised and therefore best aligned with the notion of self-supervised learning – our focus of research.

With FH regions, we show in Tab. 1e the results of our full pre-training pipeline, R-MAE, by integrating RAE and MAE. Specifically, we jointly optimize masked region autoencoding with pixel reconstruction from MAE in R-MAE. The asymmetric design that only feeds representation from pixels to regions ($\leftarrow$) achieves the best results compared to joint ($\leftrightarrow$) and regions to pixels ($\rightarrow$). Thanks to the lightweight length variant of our RAE, the improvement comes with very *minor* computational costs: the region branch only adds ~1% FLOPs to the MAE baseline (9.8b *vs*. 9.7b).[2]

**Mask ratios.** We study mask ratio as the most important hyper-parameter from MAE in Figs. 4a and 4b. Starting from the default value 0.75, we either vary the region ratio alone, or jointly with the image one. In both cases, we share the random masks whenever possible (among the image and its regions) to minimize the information leak. The results suggest that a high mask ratio (~0.75) is still required.

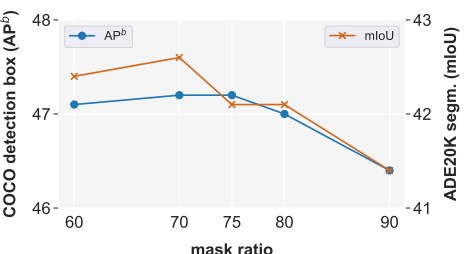

(a) **Change region mask ratio only.**

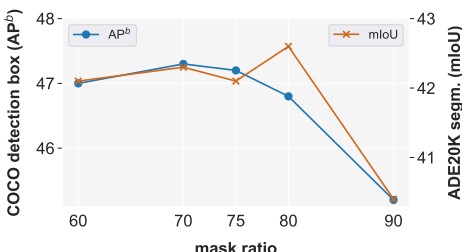

(b) **Jointly change region and image mask ratio.**

Figure 4: **Masking strategy in R-MAE.** Mask ratio matters – we either change the region mask ratio ($\beta_R$) alone (above), or jointly change it with the image mask ratio ($\beta_R = \beta_I$, bottom). In both cases, a high mask ratio (~0.75) is required.

Next, we generalize our finding and show that R-MAE performs well with high-quality regions from SAM, with more pre-training data, and on the long-tailed object detection task.

| method | $AP^b$ | $AP^m$ | mIoU |
|--------|--------|--------|------|
| MAE | 50.1 | 44.6 | 45.9 |
| R-MAE | **51.4** | **45.9** | **47.1** |

Table 2: **Pre-training on high-quality regions** generated by SAM (Kirillov et al., 2023). Similar to RAE, R-MAE pre-trained with SAM regions on COCO images outperforms MAE, showing the effectiveness of learning from regions.

**Pre-training on high-quality regions.** We show in Tab. 1d that RAE alone is an effective task when provided high-quality regions. Similarly, when integrated with MAE, R-MAE demonstrates the same behaviour as shown in Tab. 2, improving over the strong baseline from MAE.

**More data on COCO.** The second generalization is on pre-training data scale – if adding more data changes our observation. To this end, we add COCO `unlabeled2017`, and again pre-train

---

[2]Technically for R-MAE, computing regions is also an overhead. Yet FH is $O(n \log (n))$ w.r.t. number of pixels (Felzenszwalb & Huttenlocher, 2004) and pre-computable using cpus. Empirically we find it's negligible.

| method | `train2017` only | | | `+unlabeled2017` | | |
|--------|------|------|------|------|------|------|
| | $AP^b$ | $AP^m$ | mIoU | $AP^b$ | $AP^m$ | mIoU |
| MAE | 50.1 | 44.6 | 45.9 | 51.5 | 45.9 | 48.4 |
| R-MAE | **50.6** | **45.0** | **46.8** | **52.1** | **46.1** | **48.7** |

Table 3: **More pre-training data on COCO** with `train2017` + `unlabeled2017` set. R-MAE continues to outperform MAE without changing any hyper-parameters.

R-MAE with FH regions for 4k epochs following Hu et al. (2022). Results are summarized in Tab. 3. Without changing any hyper-parameters, R-MAE continues to outperform MAE.

| method | $AP^b$ | $AP^b_{rare}$ | $AP^m$ | $AP^m_{rare}$ |
|--------|------|------|------|------|
| MAE | 37.7 | 25.4 | 35.8 | 25.1 |
| R-MAE | **38.3** | **26.7** | **36.2** | **26.4** |

Table 4: **Comparison on LVIS detection** between MAE and R-MAE with FH regions. We include LVIS-specific metrics for long-tail recognition ($AP_{rare}$). The consistent improvement is observed especially for *rare* objects.

**Comparison on LVIS detection.** As a generalization of the downstream task, we further report the evaluation of R-MAE and MAE on the LVIS benchmark (Gupta et al., 2019). This dataset includes ~2m high-quality instance segmentation labels for 1203 categories that exhibit a natural, long-tailed object distribution. Unlike COCO, there is a significant imbalance in class distribution with many *rare* classes having very few (*e.g.*, <10) training examples. We use the same training recipe as Li et al. (2022b) for LVIS. We directly evaluate the backbones pre-trained with FH regions on COCO `train2017`. The results are presented in Tab. 4, where we observe a similar gain as on COCO detection. Notably, R-MAE shows a bigger improvement on the *rare*, or tail classes, suggesting the priors learned in R-MAE is more decoupled from category labels.

| # ep | method | FLOPs | $AP^b$ | $AP^m$ | mIoU |
|------|--------|-------|------|------|------|
| | SemMAE | 4.3× | - | - | 46.3 |
| 800 | MixedAE | 2.6× | 50.3 | 43.5 | **48.7** |
| | R-MAE | 1× | **51.3** | **45.7** | 46.6 |
| | MultiMAE | 2.5× | - | - | 46.2 |
| | LoMaR | 1.8× | 51.4 | 45.7 | - |
| 1600 | MixedAE | 2.6× | 51.5 | 44.5 | **49.8** |
| | Long-Seq MAE | 4.3× | 52.1 | 46.2 | - |
| | R-MAE | 1× | **52.3** | **46.4** | 47.5 |

Table 5: **State-of-the-art comparison** with ImageNet pre-training among MAE variants. FLOPs for each method is reported as relative to R-MAE.

| method | $AP^b$ | $AP^m$ | mIoU |
|--------|------|------|------|
| supervised | 47.9 | 42.9 | 47.4 |
| MoCo v3 | 47.9 | 42.7 | 47.3 |
| BEiT | 49.8 | 44.4 | 47.1 |
| R-MAE | **52.3** | **46.4** | **47.5** |

Table 6: Comparison with **other** pre-training methods.

**State-of-the-art comparison with ImageNet pre-training.** In Tab. 5 we summarize our comparison among latest MAE variants (Hu et al., 2022; Li et al., 2022a; Chen et al., 2022a; Bachmann et al., 2022; Chen et al., 2023) on COCO object detection and instance segmentation, along with ADE20K semantic segmentation. The transferring recipe follows ViTDet (Li et al., 2022b) for COCO object detection and instance segmentation (*i.e.*, 100 epochs with batch size of 64), and MAE (He et al., 2022) for ADE20K semantic segmentation (*i.e.*, 100 epochs with batch size of 16). All methods are pre-trained on ImageNet (Chen et al., 2020b; He et al., 2022).

R-MAE is the most efficient among all MAE variants in terms of computation in FLOPs. For example, Long-Seq MAE (Hu et al., 2022) and SemMAE (Li et al., 2022a) are more than 4× as expensive due to a longer sequence length.

It should also be noted that MultiMAE (Bachmann et al., 2022) employs regions extracted from a state-of-the-art detector (*i.e.*, Mask2Former (Cheng et al., 2021)) and SemMAE (Li et al., 2022a) utilizes regions generated by a variant of iBot (Zhou et al., 2022). In contrast, R-MAE simply learns to reconstruct FH regions which can be generated by an efficient clustering algorithm.

Across all the methods compared, R-MAE achieves the best results on object detection and instance segmentation. For semantic segmentation, it comes as a second, only behind the most recent MixedAE work (Chen et al., 2023) which is more expensive in compute.

To complete our picture for comparison, we also included results with other types of ImageNet-based pre-training in Tab. 6. This incudes supervised learning with labels, contrastive learning (Chen et al., 2021), and masked token prediction (Bao et al., 2022). We outperform on all the benchmarks.

**Qualitative results.** Fig. 5 shows the region reconstruction of R-MAE pre-trained with FH regions.

**R-MAE for interactive segmentation.** Since the pre-training task is to complete regions, our pre-trained R-MAE model can naturally act as 'interactive segmenter' (Sofiiuk et al., 2020). In fact,

Image      GT region      Image      FH region

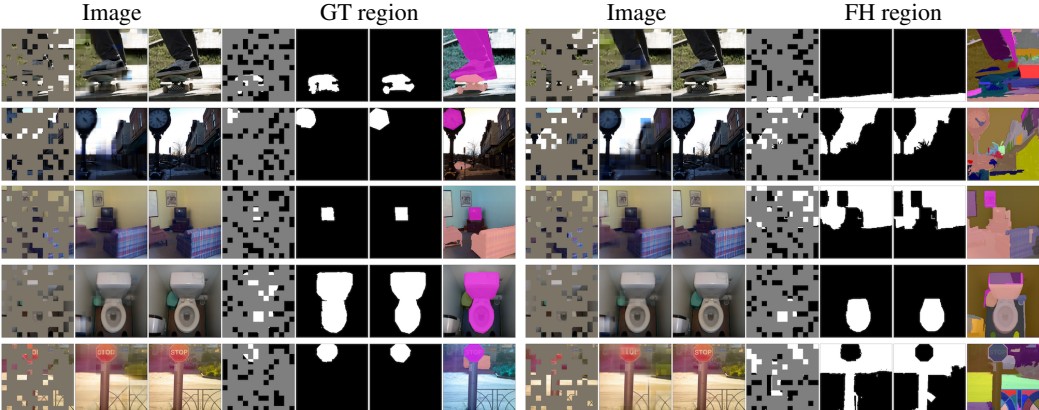

Figure 5: **Qualitative results** on COCO `val2017` images, using R-MAE pre-trained with unsupervised region maps (Felzenszwalb & Huttenlocher, 2004), and then applied on either COCO ground-truth regions (left column) or FH regions used during pre-training (right column). The image group contains 1) the masked image, 2) the image reconstruction, 3) the original image. The region group has 1) the masked region, 2) the region reconstruction, 3) the original region, 4) regions in the corresponding image. Besides results, the figure also gives a sense of the differences between ground-truths and regions used in R-MAE. Surprisingly, the algorithm pre-trained with FH regions can generalize well to ground-truth ones.

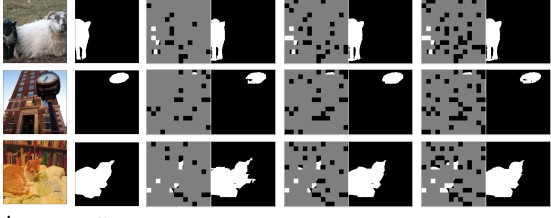

image    GT    mask 90%    mask 85%    mask 80%

Figure 6: **Interactive segmentation with R-MAE.** Here we show R-MAE's region predictions on COCO `val2017` set, given images and only masked region maps severing as a proxy to a potential user's input. Going from left to right, the user prompts more information. The model is pre-trained with a fixed region masking ratio (75%) but generates high-quality masks even with significantly higher masking ratio (90%).

if we view the visible foreground/background patches as *prompts* (Kirillov et al., 2023), then RAE shares the same task nature as SAM. While our focus is on representation learning, not on generation quality – which leads to distinctions in design (*e.g.*, efficient decoding of multiple regions with the length variant), R-MAE can still perform interactive segmentation, which we show next.

Specifically, we ask it to take the image along with some patches-of-interest as its inputs after pre-training. In an interactive segmentation setting, these patches can be provided by user clicks or eye gazing. A reasonable model can then predict the object corresponding to the given patches. From Fig. 6, we can see that the pre-trained model can indeed predict high-quality regions even with 90% of the patches masked, and continue to refine when more hints are supplied (from left to right).

## 5 CONCLUSION

In this work, we presented a simple yet effective approach (R-MAE) to explore an important vision concept – *region* in MAE (He et al., 2022). Through quantitative and qualitative results, we showed R-MAE is indeed more 'region-aware', and can consistently help downstream performance on localization-related tasks (*e.g.* detection and segmentation).

**Limitations.** While regions share resemblances to words (*e.g.*, in being discrete), there are other aspects of words that regions may still lack (*e.g.*, it's debatable if they provide enough semantics). Therefore, our work is still a first step towards truly closing the gap to words for large language models in NLP. Nevertheless, we believe our exploration is valuable towards uncovering the visual analogue of words in computer vision, and can inspire more future efforts along this direction.

While regions from SAM (Kirillov et al., 2023) significantly boost the performance of R-MAE, SAM itself initializes from MAE, is computationally expensive, and requires large-scale learning with human in the loop. A possible next step is to nail down the true reason why SAM regions are helpful, and minimize the complexities in this pipeline.

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

## A  IMPLEMENTATION DETAILS OF R-MAE

**Masking strategy.** Different from Li et al. (2022a) which deploys a biased sampling strategy using semantic parts, we aim to verify the effectiveness of R-MAE without changing the distribution of masked images. Therefore, during the pre-training stage, we simply follow the random uniform masking strategy as used in MAE (He et al., 2022). To ensure the task on the region side is meaningful, we first sample the mask applied to the image, then sample from region maps that have *at least* one visible foreground patch.

To best describe our implemented model for masked region autoencoding (RAE) and the final R-MAE framework, we resort to a more mathematical formulation of the problem and our solutions below.

**Basic notations.** We denote $R \in \mathbb{R}^{H \times W \times k}$ as the region maps corresponding to the input image, where $k$ is the number of regions, and $H, W$ are the dimensions of the input. Our model first patchifies $R$, and then masks $R$ with a ratio of $\beta_R$. The patch size $p$ used in the regions is the same as the input image. The full sequence length is denoted by $N = \frac{H}{p} \cdot \frac{W}{p}$.

**RAE channel variant.** Here, we merge $k$ region maps in the *channel* dimension, resulting in an input sequence of visible patches $v_R \in \mathbb{R}^{N \cdot (1-\beta_R) \times (k \cdot p \cdot p)}$. This can be seen as converting region maps $R \in \mathbb{R}^{H \times W \times k}$ into an image of $k$ channels. The region encoder takes $v_R$ as its input to generate region embeddings:

$$v_{\text{renc}} = \text{R–Encoder}(v_R), \tag{1}$$

where $v_{\text{renc}} \in \mathbb{R}^{N \cdot (1-\beta_R) \times p_E}$ is the region encoder's output.

We then add image features from the pixel encoder to the region embeddings from the region encoder. The augmented visual features are passed into the region decoder in order to make predictions for masked region patches:

$$v'_{\text{renc}} = \text{MaskFill}\left(f(v_{\text{renc}}), \texttt{[mask]}\right), \tag{2}$$

$$v_{\text{rdec}} = \text{R–Decoder}\left(v'_{\text{renc}} + v'_{\text{penc}}\right), \tag{3}$$

where $v'_{\text{renc}} \in \mathbb{R}^{N \times p_D}$ is the region embeddings filled with the $\texttt{[mask]}$ token and $v_{\text{rdec}} \in \mathbb{R}^{N \times p_D}$ is the output of the region decoder.

By treating $R$ as an image of $k$ channels, the channel variant demonstrates great efficiency during the pre-training process. This variant, however, fails to deal with the permutation equivariance between $k$ regions – the shuffling of the outputs is *not* guaranteed given shuffled inputs.

**RAE batch variant.** The RAE batch variant processes each region independently in the *batch* dimension. Note that the image features are shared among all $k$ different regions.

Given $R = \{R_i\}_{i=1}^{k}, R_i \in \mathbb{R}^{H \times W}$, our region encoder projects each visible patch of $R_i$ into a region embedding:

$$v_{\text{renc}_i} = \text{R–Encoder}(v_{R_i}), \tag{4}$$

where $v_{R_i} \in \mathbb{R}^{N \cdot (1-\beta_R) \times (p \cdot p)}$ are visible patches of $R_i$, and $v_{\text{renc}_i} \in \mathbb{R}^{N \cdot (1-\beta_R) \times p_E}$ is the output of the region encoder.

We then take the sum of the image features $v'_{\text{penc}}$ and $v'_{\text{renc}_i}$, and feed it to the region decoder for prediction:

$$v'_{\text{renc}_i} = \text{MaskFill}\left(f(v_{\text{renc}_i}), \texttt{[mask]}\right), \tag{5}$$

$$v_{\text{rdec}_i} = \text{R–Decoder}\left(v'_{\text{renc}_i} + v'_{\text{penc}}\right), \tag{6}$$

where $v'_{\text{penc}} \in \mathbb{R}^{N \times p_D}$ is the image features from the pixel encoder filled with $\texttt{[mask]}$ token. Similarly, $v'_{\text{renc}_i} \in \mathbb{R}^{N \times p_D}$ is region embeddings filled with the $\texttt{[mask]}$ token. Here, $f : p_E \to p_D$ denotes the linear projection and $v_{\text{rdec}_i} \in \mathbb{R}^{N \times p_D}$ is the region decoder output which is then used to predict masked patches of $R_i$.

While preserving the *permutation equivariance*[3] of $k$ region maps, the RAE batch variant can be computationally expensive and resource-intensive (*i.e.*, the total number of FLOPs increases linearly w.r.t. $k$).

---

[3] If one permutes the order for the $k$ input regions, the output will be shuffled in the exactly same order.

| Query | MoCo v3 | MAE | R-MAE | Query | MoCo v3 | MAE | R-MAE |
|---|---|---|---|---|---|---|---|

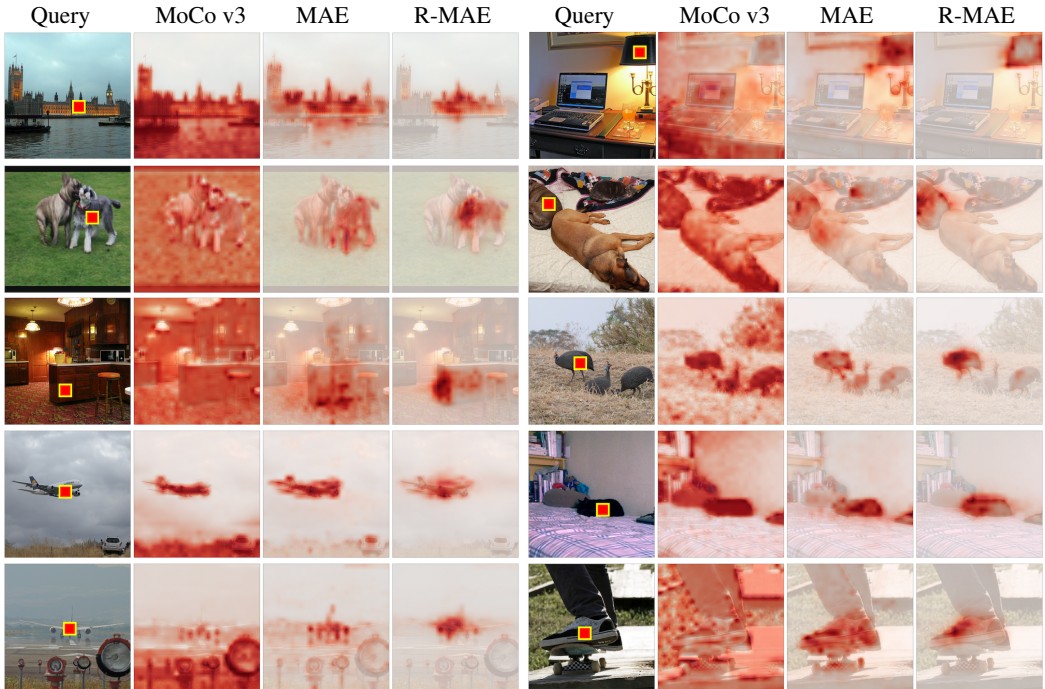

Figure 7: **Additional visualizations of attention maps** on COCO `val2017`. In each group from left to right we show the original image with the selected query (denoted by red square); three attention maps corresponding to the query generated from i) MoCo v3 (Chen et al., 2021); ii) MAE (He et al., 2022); and iii) R-MAE; all pre-trained on COCO `train2017`. Darker red colors in the attention map have larger attention weights.

**RAE length variant.** Inspired by the design of object queries in the DETR decoder (Carion et al., 2020), the RAE length variant encodes each region map into a single vector using region encoder. The region queries will be concatenated along the sequence *length* dimension as follows:

$$v_{\text{renc}_i} = \text{AvgPool}\left(\text{R–Encoder}(v_{R_i})\right), \tag{7}$$

$$v_{\text{emb}} = \text{Concat}(v_{\text{renc}_1}, ..., v_{\text{renc}_k}), \tag{8}$$

where $v_{R_i} \in \mathbb{R}^{N \cdot (1 - \beta_{\text{R}}) \times (p \cdot p)}$ are visible patches of $R_i$, $v_{\text{renc}_i} \in \mathbb{R}^{p_{\text{E}}}$ is the region embedding of $i$-th region, $v_{\text{emb}} \in \mathbb{R}^{k \times p_{\text{E}}}$ denotes the region queries, and $\text{AvgPool}$ is the average pooling operation.

Different from the pixel decoder, the region decoder contains three sub-layers in each block: self-attention, cross-attention, and feed-forward (Vaswani et al., 2017). In addition, we use a Neck module to provide cross-attention with information from pixels as context. The blocks in Neck share the same design as the ones in the pixel decoder:

$$v_{\text{context}} = \text{Neck}(v'_{\text{penc}}), \tag{9}$$

where $v'_{\text{penc}}$ is the image features filled with [mask] tokens and $v_{\text{context}} \in \mathbb{R}^{N \times p_{\text{D}}}$ is the output of Neck. The region decoder then decodes region queries with context information:

$$v_{\text{query}} = \text{R–Decoder}(f(v_{\text{emb}}), v_{\text{context}}), \tag{10}$$

where $v_{\text{query}} \in \mathbb{R}^{k \times p_{\text{D}}}$ is the output of the query decoder. Since masked region autoencoding predicts $R \in \mathbb{R}^{k \times H \times W}$ during the pre-training, we modify the cross-attention sub-layer of the last region decoder layer to expand each region embedding in $v_{\text{query}}$ into a region map as follow (see Fig. 2):

$$v_{\text{rdec}} = W^{\top} v_{\text{context}} + v_{\text{query}}[:, \text{None}], \tag{11}$$

where $W \in \mathbb{R}^{p_{\text{D}} \times p_{\text{D}}}$ is a learnable weight, $v_{\text{query}}[:, \text{None}] \in \mathbb{R}^{k \times 1 \times p_{\text{D}}}$[4], and $v_{\text{rdec}} \in \mathbb{R}^{k \times N \times p_{\text{D}}}$. The expansion in our cross-attention sub-layer can be viewed as the attention operation on each feature vector of $v_{\text{context}}$ (*i.e.*, the attention score of a single feature over itself is equal to 1). A 3-layer MLP

Image      GT region      Image      FH region

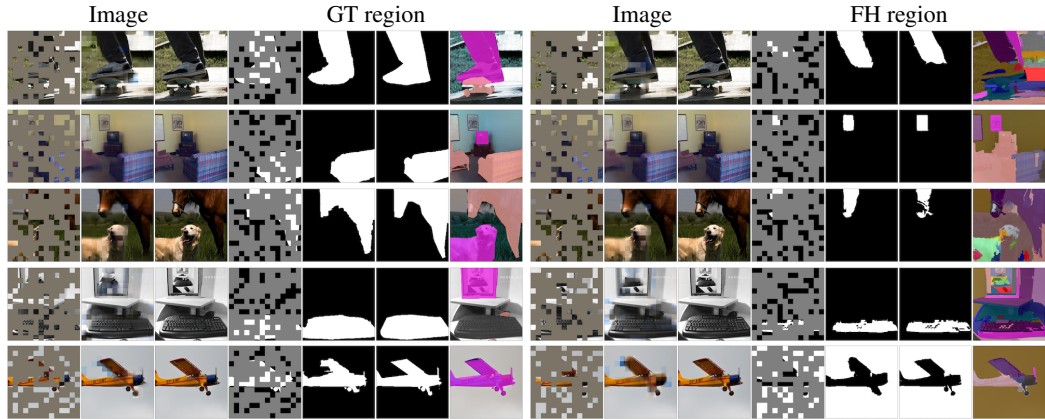

Figure 8: **Additional qualitative results** on COCO `val2017` using R-MAE pre-trained with unsupervised region maps (Felzenszwalb & Huttenlocher, 2004), and then applied on either COCO ground-truth regions (left column) or FH regions used during pre-training (right column). The image group contains 1) the masked image, 2) the image reconstruction, 3) the original image. The region group has 1) the masked region, 2) the region reconstruction, 3) the original region, 4) regions in the corresponding image.

projection, $g : \mathbb{R}^{p_D} \to \mathbb{R}^{p \cdot p}$, is then applied onto $v_{\text{rdec}}$ with a binary cross entropy loss to reconstruct $R$.

**Cross-feeding.** Let $v'_{\text{penc}}$ denotes the output of the pixel encoder filled with `[mask]` token and $v'_{\text{renc}}$ denotes the output of the region encoder filled with `[mask]` token before the pooling function. We examine three different cross-feeding styles between regions and pixels: RAE $\to$ MAE (region-to-pixel), RAE $\leftarrow$ MAE (pixel-to-region), and RAE $\leftrightarrow$ MAE (bidirectional). The default design in R-MAE follows RAE $\leftarrow$ MAE (*e.g.* see Fig. 1), and we detail the other two below.

In RAE $\to$ MAE (region-to-pixel), we add region features to the pixel features and feed it as the input to the pixel decoder in order to regress the masked image patches:

$$v'_{\text{renc}_i} = \text{MaskFill}\left(h(v_{\text{renc}_i}), \text{[mask]}\right), \tag{12}$$

$$v'_{\text{renc}} = \text{Concat}(v'_{\text{renc}_1}, ..., v'_{\text{renc}_k}), \tag{13}$$

$$v_{\text{pdec}} = \text{P--Decoder}\left(v'_{\text{penc}} + \text{AvgPool}\left(v'_{\text{renc}}\right)\right), \tag{14}$$

where $d_D$ is the dimension of pixel decoder, $v'_{\text{renc}_i} \in \mathbb{R}^{N \times d_D}$ is the output of the region encoder filled with `[mask]` token for $i$-th region, and $h : p_E \to d_D$ is a linear projection layer.

RAE $\leftrightarrow$ MAE (bidirectional) feed in both directions.

## B  MORE COMPARISONS OF OUR MAE VS. R-MAE

| pre-train | COCO | | COCO++ | | ImageNet | |
|---|---|---|---|---|---|---|
| learning rate | AP$^b$ | AP$^m$ | AP$^b$ | AP$^m$ | AP$^b$ | AP$^m$ |
| MAE w/ 1.5e-4 | 49.9 | 44.4 | 51.6 | 45.7 | 51.6 | 45.9 |
| MAE w/ 1e-4 | 50.1 | 44.6 | 51.5 | 45.9 | 51.8 | 46.1 |

Table 7: MAE with **different base learning rates**. For ImageNet w/ 1.5e-4, we directly cite the results from ViTDet (Li et al., 2022b), while others are from our own experiments. Our default setting (w/ 1e-4), chosen due to better stability, can reproduce *all* MAE results.

**MAE baselines.** We first show the comparison of MAE with different base learning rates: 1.5e-4 in (He et al., 2022) and 1e-4 in our study. Here, models are pre-trained either on ImageNet (Deng et al., 2009) with 1600 epochs, or on COCO (`train2017`)/COCO++ (`train2017` +

---

[4][:, None] indicates the dimension expansion of $v_{\text{query}}$, as in numpy.

`unlabeled2017`) with 4k epochs. All other settings are set as default. Tab. 7 shows that MAE with 1e-4 rate is able to reproduce ViTDet (Li et al., 2022b). The only reason for this change is better pre-training stability which allows us to incorporate additional loss from the masked region autoencoding. Our R-MAE shows further improvements beyond Tab. 7.

| pre-train settings | region | FLOPs | AP$^{b}$ | AP$^{m}$ | mIoU |
|---|---|---|---|---|---|
| MAE | - | 9.7b | 50.1 | 44.6 | 45.9 |
| RAE, default | FH | 4.7b | 47.2 | 41.8 | 42.1 |
| RAE, $p_{D}$=256 | | 4.8b | 47.6 | 42.2 | 42.9 |
| RAE, $p_{D}$=256 | SAM | 4.8b | 49.9 | 44.2 | 46.0 |
| RAE, $p_{D}$=256, $\beta_{I}=\beta_{R}$=.6 | | 7.3b | **50.6** | **45.1** | **46.8** |

Table 8: Exploring **better regions** from SAM (Kirillov et al., 2023) to validate RAE. We simply swap FH regions with off-the-shelf SAM ones, and with a larger decoder and changes in mask ratios, we find RAE alone can achieve better results with less compute.

**Better regions.** To further validate the design of masked region autoencoding (RAE), we explore better regions generated by an off-the-shelf segmentation model from SAM (Kirillov et al., 2023) to replace FH. With a larger region decoder and mask ratio 60%, RAE alone can achieve *better* results than MAE with *less* compute in FLOPs as shown in Tab. 8. Interestingly, we find that RAE with high-quality regions from SAM benefits from a masking ratio of 60%. We hypothesize that SAM regions contain highly semantic information that is of less redundancy and therefore require a lower masking ratio (*i.e.*, the masked language modeling only predicts a few missing words ~15%).

| pre-train | ViT-Base | | | | ViT-Large | | | |
|---|---|---|---|---|---|---|---|---|
| | AP$^{b}$ | AP$^{m}$ | mIoU | FLOPs | AP$^{b}$ | AP$^{m}$ | mIoU | FLOPs |
| MAE | 51.8 | 46.1 | **47.9** | 9.7b | 55.6 | 49.3 | 52.3 | 20.6b |
| R-MAE | **52.3** | **46.4** | 47.5 | 9.8b | **55.8** | **49.7** | **52.5** | 20.7b |

Table 9: **Larger backbones** pre-trained on ImageNet. Here, R-MAE is pre-trained to reconstruct FH regions. The gains from R-MAE can hold despite less relative computation overheads of only 0.5%.

**Larger backbones.** Tab. 9 shows the scaling trend of model size when pre-trained on ImageNet. Overall, the gains can hold at ViT-L (Dosovitskiy et al., 2020), despite even more negligible computational overheads of only 0.5% with larger backbones.

| pre-train | fine-tune | | linear-eval | |
|---|---|---|---|---|
| | Acc@1 | Acc@5 | Acc@1 | Acc@5 |
| MAE | 83.6 | 96.6 | 68.0 | 87.3 |
| R-MAE | 83.6 | 96.6 | 60.6 | 82.4 |

Table 10: **ImageNet classification** as downstream task for MAE and R-MAE. The representation from R-MAE is more locally focused and less fit for linear-eval, but fine-tuning fixes the gap.

**ImageNet classification.** To give a more complete assessment, we also evaluate our pre-trained models on ImageNet classification. To be consistent with MAE (He et al., 2022), we pre-train the ViT with R-MAE on ImageNet for 1600 epochs. It can be seen from Tab. 10 that our R-MAE achieves the same performance with MAE when being fine-tuned end-to-end. Interestingly, the linear probing performance of R-MAE lags behind MAE by a large margin. This observation indicates that our R-MAE is more focused on local patterns rather than global average features suited for image classification.

| pre-train | accuracy |
|---|---|
| MAE | 95.8 |
| R-MAE | 96.2 |

Table 11: **Fine-grained classification** using Flower dataset (Nilsback & Zisserman, 2008) as downstream task for MAE and R-MAE. The representation from R-MAE is better on capturing fine-grained details.

**Fine-grained classification.** As the representation of R-MAE is more locally focused, we evaluate both models pre-trained with R-MAE and MAE on fine-grained classification using Flower dataset Nilsback & Zisserman (2008). Both models are pre-trained on ImageNet for 1600 epochs. Tab. 11 shows that R-MAE outperforms MAE in this task which again confirms the effectiveness of R-MAE in learning local representation.

## C ADDITIONAL VISUALIZATIONS

We provide extra qualitative results of our pre-trained models in Fig. 7 and Fig. 8.

## D ASSET LICENSES

| Dataset | License |
| --- | --- |
| ImageNet (Deng et al., 2009) | https://image-net.org/download.php |
| COCO (Lin et al., 2014) | Creative Commons Attribution 4.0 License |
| ADE20K (Zhou et al., 2019) | Creative Commons BSD-3 License Agreement |
| LVIS (Gupta et al., 2019) | Creative Commons Attribution 4.0 License |
| Oxford 102 Flower (Nilsback & Zisserman, 2008) | MIT License |

