# OpenReview forum: "R-MAE: Regions Meet Masked Autoencoders"
_ICLR.cc/2024/Conference — ICLR 2024 poster_

### Official Review · Reviewer_FunS · 2023-10-25

**Soundness:** 2 fair
**Presentation:** 4 excellent
**Contribution:** 2 fair
**Rating:** 6
**Confidence:** 5

**Summary:**

The paper studies regions in masked autoencoders. The idea is interesting although there has been a large amount of self-supervised learning introducing the concept of regions (These methods are generally called self-supervised object detection methods in the field). The resulting R-MAE shows better performance on COCO, ADE20K datasets compared with MAE baseline.

**Strengths:**

1. The overall paper is clear and easy to follow.
2. The analysis for different designs of regions is comprehensive.
3. This paper gives some suggestions when training with regions in MIM.
4. The authors will open the source code and models.

**Weaknesses:**

1.	Self-supervised contrastive learning needs to introduce the concept of region to focus local information due to its a priori assumption of image semantic consistency, but MAE does not have this problem. Moreover, I agree that the reconstruction of raw pixel values lacks a higher level of semantic information for image understanding compared to word reconstruction in NLP. However, I do not agree the introduction of binary regions adds high-level semantics. Therefore, I argue this paper is the same as previous self-supervised object detection learning. The effect comes from further learning of the local region, so it is effective in tasks such as detection and segmentation. At the same time, the performance of this type of method will be lower than the baseline on tasks such as ImageNet image classification. If high-level semantic understanding tasks such as image classification do not perform well, then it is difficult to say that R-MAE has a high-level understanding as mentioned in the introduction.
2.	Like question 1, the paper lacks ImageNet classification experiments.
3.	The calculation amount comparison is unfair. The paper only calculates the calculation amount of the architecture, but the results of the FH algorithm and SAM region generation are not included.
4.	From Table 3, more data show that the gain in segmentation results relative to MAE is weaker. Is there no difference between the results of MAE and R-MAE on a larger data set?

**Questions:**

Please refer to weaknesses.

---

> ### Author Response · Authors · 2023-11-13
> **Author rebuttal**
>
> 1. ***Self-supervised contrastive learning needs to introduce the concept of region to focus local information due to its a priori assumption of image semantic consistency, but MAE does not have this problem. Moreover, I agree that the reconstruction of raw pixel values lacks a higher level of semantic information for image understanding compared to word reconstruction in NLP. However, I do not agree that the introduction of binary regions adds high-level semantics. Therefore, I argue this paper is the same as previous self-supervised object detection learning. The effect comes from further learning of the local region, so it is effective in tasks such as detection and segmentation.***
>
> - Thanks for the feedback. While we indeed wanted to pursue the visual analogue of words as our goal, we also understand the concern that regions do not automatically translate to high-level semantic information. On the other hand, regions do offer resemblances to words that encode *discrete* information about which set of pixels should be in the same “group”, and we believe it’s a better way to phrase it. We have updated our introduction to reflect this, and added another limitation to reduce confusion. Please see page 1 and page 9 of the latest draft.
>
> 2. ***Like question 1, the paper lacks ImageNet classification experiments.***
>
> - We have included ImageNet classification results as part of our supplementary material, we make a copy below for convenience:
>
> |pre-train|Finetune - Acc@1|Finetune - Acc@5|Linear - Acc@1|Linear - Acc@5
> |--|:--:|:--:|:--:|:--:
> MAE|83.6|96.6|68.0|87.3
> R-MAE|83.6|96.6|60.6|82.4
>
> - In the fine-tuning setting, we find MAE and R-MAE to be at-par. For linear probing, we do find MAE to be more effective.
> - Nevertheless, we are no longer claiming regions introduce semantics into MAE pre-training, and believe our exploration and discovery are still valuable.
>
> 3. ***The calculation amount comparison is unfair. The paper only calculates the calculation amount of the architecture, but the results of the FH algorithm and SAM region generation are not included.***
>
> - Thanks for pointing this out. We have added a footnote on page 7 to address this concern. Indeed, R-MAE needs extra overhead to compute regions. Yet FH [R1] is designed to be highly efficient and has a complexity of O(n log n) where n is the number of pixels. Moreover, regions are only computed once per image and can be pre-computed with CPUs. In practice, we do not observe any overhead when loading it from the disk.
>
> - As for SAM, it’s indeed expensive (added a note on page 9), but its main purpose in our paper is to serve as an oracle that provides upper bounds. Following FH, we also pre-computed SAM regions and loaded them during pre-training. We look forward to more studies on making it more efficient.
>
> [R1] Felzenszwalb, Pedro F., and Daniel P. Huttenlocher. "Efficient graph-based image segmentation." International journal of computer vision 59 (2004): 167-181.
>
> 4. ***From Table 3, more data show that the gain in segmentation results relative to MAE is weaker. Is there no difference between the results of MAE and R-MAE on a larger data set?***
>
> - For mIoU, semantic segmentation on ADE20K is in general less reliable as a trend indicator. For example, see Fig. 4a and Fig. 4b as we reported both AP$_b$ and mIoU. While the trend in AP$_b$ is smooth, in mIoU there are outliers even though we average over multiple seeds for each point on the plot.  We hypothesize it’s due to the size of the dataset.
> - For AP$_m$, we also reported pre-training on ImageNet which is 10x larger, and still observed healthy gains over MAE, copied below. The gain on AP$_m$ is increased from COCO++ to ImageNet from 0.2 to 0.3. Here, R-MAE are pre-trained with FH regions:
>
> |pre-train|COCO - AP$_b$|COCO - AP$_m$|COCO++ - AP$_b$|COCO++ - AP$_m$|ImageNet - AP$_b$|ImageNet - AP$_m$
> |--|:--:|:--:|:--:|:--:|:--:|:--:
> MAE|50.1|44.6|51.5|45.9|51.8|46.1
> R-MAE|50.6 (+0.5)|45.0 (+0.4)|52.1 (+0.6)|46.1 (+0.2)|52.3 (+0.5)|46.4 (+0.3)
>
> Here, COCO means COCO train set and COCO++ means COCO train + unlabelled set
>
> - In general, we believe the trend is also dependent on other factors (e.g., types of data, types of masks used given the strong results with SAM regions), and may not be simply concluded from results of a single experiment.

---

### Official Review · Reviewer_uFze · 2023-10-30

**Soundness:** 2 fair
**Presentation:** 3 good
**Contribution:** 2 fair
**Rating:** 6
**Confidence:** 5

**Summary:**

This paper proposes a new pretext task called mask Region Autoencoding for self-supervised visual representation learning. Instead of considering pixels as the operated units, the authors conduct masking and reconstruction on the so-called region levels. By integrating RAE into MAE, the resulting R-MAE achieves impressive performance on various transfer learning settings.

**Strengths:**

- The paper is generally well-written with solid experimental results.
- The paper has a clear explanation of the proposed method with valid qualitative demonstration.
- Beside common transfer learning experiments, the authors also explore the usage of R-MAE for interactive segmentation.

**Weaknesses:**

- About the importance of regions:
  - As discussed in Sec. 2, the authors claim that there are many different sources to obtain regions. In other words, here regions do not have a specific definition, especially under the context of unsupervised learning. Or in another words, the best definition of regions might differ with respect to different downstream tasks, while just for object detection and semantic segmentation, SAM proposals might be the best.
  - One following question to explain is that why in SAM as the region source can even perform better than panoptic ground truth COCO annotations when pre-trained on COCO, as in Tab. 1(d)?
  - Also, as claimed in 3rd paragraph of Sec. 1, the authors claim that "regions" might perform similarly with "words" in language models to improve the scalability of MAE and pursue visual emergent properties. Unfortunately, both the qualitative and quantitative results can only demonstrate similar observation for the learned representation with MAE. Moreover, the authors only conduct experiments with ViT-B, without further exploration about the scalability of R-MAE.
  - Therefore, it is hard to convince me that this work has a different motivation with locality reconstruction works like LoMaR [1] and SemMAE.
- About the architecture:
  - Does the region encoder share weights with the pixel encoder? If not, which one would be transferred for downstream tasks in the context of RAE and R-MAE respectively?
  - Is there any advantage to conduct region reconstruction only for binary region masks, which totally throw the RGB information, while the latter should also be part of the semantics? This question is way more interesting if we consider that RAE with SAM performs better than MAE.
  - One following question would be what will happen if we transfer RAE and MAE weights to downstream color-sensitive tasks, like the the Flowers classification dataset.
  - Moreover, is there any advantage to maintain a separate branch of region encoder, since a simpler implementation might be similar with LoMaR, where for the visible part of each region, we can directly utilize them to reconstruct the RGB values of the masked part of this specific region, so that we can perform the two objectives of R-MAE at the same with without introducing a separate branch.
  - Just to make sure I have understood correctly, does RAE mean we only apply the upper part (=region encoder + region decoder) of Fig. 1?
- Overall, it is hard to convince me that region modeling is so important as the authors claim and it seems like there are much easier ways to implement this idea than the proposed R-MAE framework.

[1] Chen, Jun, et al. "Efficient self-supervised vision pretraining with local masked reconstruction." *arXiv preprint arXiv:2206.00790* (2022).

[2] Chen, Kai, et al. "Mixed autoencoder for self-supervised visual representation learning." *Proceedings of the IEEE/CVF Conference on Computer Vision and Pattern Recognition*. 2023.

[3] Liu, Jihao, et al. "Mixmim: Mixed and masked image modeling for efficient visual representation learning." *arXiv preprint arXiv:2205.13137* (2022).

**Questions:**

- About experiments:
  - R-MAE has been surpassed by earlier MAE-based framework targeting at detection and segmentation with local awareness (e.g., 800-epoch MixedAE [2] outperforms 1600-epoch R-MAE on ADE20K).
  - It would be better to also report quantitative comparison for interactive segmentation for better understanding in Fig. 6.

[1] Chen, Jun, et al. "Efficient self-supervised vision pretraining with local masked reconstruction." *arXiv preprint arXiv:2206.00790* (2022).

[2] Chen, Kai, et al. "Mixed autoencoder for self-supervised visual representation learning." *Proceedings of the IEEE/CVF Conference on Computer Vision and Pattern Recognition*. 2023.

[3] Liu, Jihao, et al. "Mixmim: Mixed and masked image modeling for efficient visual representation learning." *arXiv preprint arXiv:2205.13137* (2022).

---

> ### Author Response · Authors · 2023-11-13
> **Author rebuttal**
>
> Since this is a multi-folded question, we address them with bullet points:
>
> 1. ***About the importance of regions:***
>
> - While regions can be generally interpreted as groups of pixels given an image, we agree a more rigorous definition would be helpful. We also fully agree that the most suitable regions can vary from task to task, but we also want to point out *the value of finding the right regions for important tasks*. For example, physiological theories [R1] suggest that human perception will group similar elements and parts together to parse complex scenes and objects. It would be of tremendous value to discover such regions, not only for the scientific understanding of intelligence, but also for practical applications that interact with humans.
> - Yes this is an intriguing empirical result. One hypothesis is that SAM covers a more diverse set of regions (e.g., not just the entire area of a human, but also parts such as faces and clothes) beyond the ~200 classes defined for panoptic segmentation; the other hypothesis is that SAM regions are more consistent and high in recall, whereas human annotations can be subject to disagreement and carelessness. As written in our last paragraph, we are also curious to test these hypotheses and simplify/accelerate our pipeline as future work.
> - Thanks for pointing this out. We provided the comparisons on ViT-L as part of our supplementary material (see Table C), which has even less *relative* extra computation from the RAE branch. However, we do agree that the tone in the introduction needs a revisit. We have made updates to the introduction and added another limitation to specifically address this concern. Please check them out.
>
> [R1] Koffka, Kurt. Principles of Gestalt psychology. Vol. 44. Routledge, 2013.
>
> 2. ***About the architecture:***
>
> - Sorry for the confusion, but no, the region encoder only operates on binary region maps and does not share weights with the pixel encoder.
> - Both RAE and R-MAE have the pixel encoder. In the case of RAE, the pixel encoder gets pre-trained from reconstructing regions as it feeds into the region decoder.
> - The pixel encoder gets transferred to downstream tasks. It has seen RGB images before and can in fact be directly used as an interactive segmenter.
> - All tasks we have experimented with are color-sensitive ones, including COCO/LVIS object detection and segmentation, and ADE20K semantic segmentation. However, we are happy to conduct more classification experiments (e.g., on Flowers) if the reviewer feels the need.
> - Thanks for the suggestion on joint-branch training. Indeed it’s a valid idea and if we understand correctly, has at least been explored in SemMAE. We compare with SemMAE in Table 5, in which we show advantage in both efficiency and effectiveness.
> - Maintaining a separate branch for the region encoder is especially helpful when there are a large number of regions per-image. The size of the region encoder can then be decoupled to the pixel encoder, and maintain *permutation equivariance* as we did in our "length-variant".
> - We also empirically find the asymmetric design that feeds pixel encoder for regions but not vice versa works best. Using region encoders to predict pixels is not as effective, as shown in Tab. 1e.
>
> 3. ***R-MAE has been surpassed by earlier MAE-based framework targeting at detection and segmentation with local awareness (e.g., 800-epoch MixedAE [2] outperforms 1600-epoch R-MAE on ADE20K).***
>
> - Thanks for the pointer. We have updated our draft and included MixedAE [2] in our comparison (Tab. 5). Please check out page 8.
> - We find MixedAE works well for semantic segmentation and has acknowledged it in the main text. However, we do also find our R-MAE results on COCO to be more competitive (according to Tab. 1 of [2]) and with *less* per-iteration compute during pre-training.
>
> 4. ***It would be better to also report quantitative comparison for interactive segmentation for better understanding in Fig. 6.***
>
> - Thanks for the suggestion. We computed the mIoU between the predicted regions from R-MAE and ground-truth regions on COCO for a subset of its validation images. The results are shown below:
>
> ||mIoU
> |--|:--:
> mask 90%|76
> mask 85%|82
> mask 80%|86
>
> - With mask ratio 90%, RAE can already reach an mIoU of 76%. Note that this is highly competitive as a promptable segmenter according to the SAM paper (Fig. 9c and Fig. 9d), where even the initial oracle is below 75%.
> - As expected, more visibility leads to better results. This is also consistent with our visualizations.

---

> > ### Comment · Reviewer_uFze · 2023-11-22
> > **Respone to Author Rebuttal**
> >
> > Thanks for the detailed rebuttal. My concerns have been largely addressed, but I still hope the authors can 1) better narrow down and explain the definiton of "regions" adopted in this paper and 2) add the Flowers experiments to support the utilization of binary masks. I would like to raise my score to 6. Hope my reviews can help the authors better revise the paper.

---

> > > ### Author Response · Authors · 2023-11-23
> > >
> > > Thanks for the timely acknowledgment and feedback to our rebuttal.
> > >
> > > In response to the suggestions: 1) we are committed revise the draft and reflect the definition of "regions" with better clarity and precision. 2) we are actively running Flowers experiments now, and while results are unlikely to be ready by the deadline (in a few hours), we will definitely include them in the revision.
> > >
> > > Again, we appreciate the feedback and suggestions in enhancing the quality of our work.

---

### Official Review · Reviewer_YqrJ · 2023-10-31

**Soundness:** 3 good
**Presentation:** 3 good
**Contribution:** 3 good
**Rating:** 6
**Confidence:** 3

**Summary:**

The authors introduce a self-supervised image representation learning method called "masked region autoencoding" (RAE), treating regions as the visual equivalent of words. When integrated with the existing Masked Autoencoding (MAE) approach, the combined method (R-MAE) consistently improves performance in various vision tasks. RAE offers a more region-aware and instance-aware representation of images.

**Strengths:**

1. The paper is highly clear in its presentation, effectively conveying the proposed methodology with its motivation.
2. The paper extends the traditional Masked Autoencoding (MAE) approach by considering regions as visual analogs of words. The concept of using regions for interactive segmentation is also original.
3. The proposed method can consistently help downstream performance on localization-related tasks (e.g., detection and segmentation).

**Weaknesses:**

1. The paper could benefit from a more extensive comparison with existing methods in the field. While it highlights the strengths of R-MAE, a more in-depth quantitative comparison with other state-of-the-art self-supervised learning techniques (based on MAE) would strengthen the paper.

**Questions:**

1. Is the performance of R-MAE sensitive to the quality of region maps, and are there strategies to mitigate this sensitivity?
2. Could the authors provide a more extensive quantitative comparison with other state-of-the-art self-supervised learning methods in computer vision? This would help readers understand how R-MAE performs in relation to existing techniques.

---

> ### Author Response · Authors · 2023-11-13
> **Author rebuttal**
>
> 1. ***Could the authors provide a more extensive quantitative comparison with other state-of-the-art self-supervised learning methods in computer vision? This would help readers understand how R-MAE performs in relation to existing techniques***
>
> - In Table 5, we have compared R-MAE to several state-of-the-art reconstructive pre-training methods when pre-trained on the standard ImageNet dataset. To make it more comprehensive, we added more results from:
>     - (1) a latest work published at CVPR 2023 (MixedAE);
>     - (2) supervised pre-training;
>     - (3) contrastive pre-training (and more) to the draft.
>
> - The results of (2) and (3) can offer a more complete picture of other self-supervised learning methods on this task. Please check page 8 of the updated draft for the results and the accompanied descriptions.
>
> 2. ***Is the performance of R-MAE sensitive to the quality of region maps, and are there strategies to mitigate this sensitivity?***
>
> - In Tab. 1d, we show results with regard to different sources of regions (to better capture the sensitivity, we conducted this analysis on RAE without the MAE branch). Several remarks:
>     - High-quality region maps indeed help a lot! With pre-computed SAM regions (as an oracle), RAE alone can achieve better results than MAE.
>     - RAE also works gracefully when the region quality degenerates, e.g., with FH.
>     - One interesting observation is that RAE favors regions that densely cover the whole image. For example, RAE does not work well for ground-truth COCO instance annotations (AP$_b$=44.7, AP$_m$=39.5, mIoU=41.6). However, dense coverage requires our model to be highly efficient with a large number of regions per-image (e.g., 54.5 on average for SAM regions). Therefore, our ```length-variant``` can be viewed as an effective mitigation strategy.
>
> - In general, R-MAE works with various types of regions from a simple graph segmentation algorithm like FH to deep learning models like SAM.

---

### Author Response · Authors · 2023-11-13
**Author rebuttal**

We would like to thank all the reviewers for spending their time and providing valuable feedback to our work.
- To recap the strengths, various aspects of our paper are recognized:
    - **Writing**: "highly clear in its presentation, effectively conveying" [```YqrJ```]; "generally well-written", "clear explanation of the proposed method" [```uFze```]; "clear and easy to follow" [```FunS```]
    - **Novelty**: "using regions for interactive segmentation is also original" [```YqrJ```]; "idea is interesting" [```FunS```]
    - **Results**: "can consistently help downstream performance" [```YqrJ```]; "achieves impressive performance", "solid experimental results", "valid qualitative demonstration" [```uFze```]; "the analysis for different designs of regions is comprehensive" [```FunS```]
    - **Reproducibility**: "will open the source code and models" [```FunS```]
- We hope to address all the remaining concerns in this rebuttal period. In addition to the detailed responses to each review, we have also **uploaded an updated PDF file** with modifications highlighted in blue. We will make references when it's best to check the specific sections in the updated draft.
- We also respond to each individual reviewer with results and explanations below. Please let us know if there is anything unclear or any remaining concern. We are happy to address them.

---

### Author Response · Authors · 2023-11-20
**We would like to hear reviewers' feedback**

Dear reviewers,

With the upcoming deadline of the reviewer-author discussion period, we extend our heartfelt gratitude for your invaluable time and expertise in assessing our paper. Your insights have contributed significantly to refining our work.

We would really appreciate the opportunity to engage in constructive dialogue to improve our research. *We eagerly welcome your acknowledgement and input if there are any more concerns or further need for clarifications*. If **additional experimental results** and **draft updates** are needed, then we do have a hard deadline to catch up within 2 days.

Looking forward!

Authors

---

### Meta-Review · Area_Chair_Gaos · 2024-01-04

**Metareview:**

This paper presents a new representation learning method, which augments MAEs to model object region masks. The method takes as input object masks from an off-the-shelf object region detector, along with the RGB image. Then method trains a variant of an MAE on these inputs, with region masks stacked in channels. The experiments show improved performance on popular benchmarks. The paper's strengths are its simplicity, performance, and clear experiments. Weaknesses include that this is not the first paper to propose region-awareness for representation learning, and some reviewers were not convinced that this is the best way to achieve region-awareness.

**Justification For Why Not Higher Score:**

Reviewers were borderline on this paper. The SAC feels the novelty and performance are not substantial enough to merit a spotlight or oral.

**Justification For Why Not Lower Score:**

All reviewers came to the consensus that the paper is marginally above the acceptance threshold. The SAC concurs.

---

### Decision · Program_Chairs · 2024-01-16

Accept (poster)